# Losses Related to Breast Cancer Diagnosis: The Impact on Grief and Depression Symptomatology Within the Context of Hispanic/Latina Patients with Breast Cancer

**DOI:** 10.3390/healthcare13060624

**Published:** 2025-03-13

**Authors:** Cristina Peña-Vargas, Paola del Río-Rodriguez, Lianel P. Rosario, Guillermo Laporte-Estela, Normarie Torres-Blasco, Zindie Rodriguez-Castro, Nelmit Tollinchi-Natali, Willa I. Guerrero, Patsy Torres, Guillermo N. Armaiz-Pena, Eida M. Castro-Figueroa

**Affiliations:** 1School of Behavioral and Brain Sciences, Ponce Health Sciences University, Ponce 00716, Puerto Rico; pdelrio22@stu.psm.edu (P.d.R.-R.); lrosario21@psm.edu (L.P.R.); glaporte22@stu.psm.edu (G.L.-E.); normarietorres@psm.edu (N.T.-B.); wguerrero23@stu.psm.edu (W.I.G.); ptorres23@stu.psm.edu (P.T.); ecastro@psm.edu (E.M.C.-F.); 2Ponce Research Institute, Ponce Health Sciences University, Ponce 00716, Puerto Rico; zrodriguez@psm.edu (Z.R.-C.); garmaiz@psm.edu (G.N.A.-P.); 3School of Dental Medicine, Ponce Health Sciences University, Ponce 00716, Puerto Rico; ntollinchi@psm.edu; 4School of Medicine, Ponce Health Sciences University, Ponce 00716, Puerto Rico

**Keywords:** cancer, loss, grief, affective neuroscience, depression, psycho-oncology

## Abstract

**Objective**: The present study explored the association between the losses incurred due to breast cancer diagnosis, symptoms of depression, PANIC/GRIEF, and contextual factors within the context of Hispanic/Latina (H/L) patients diagnosed with breast cancer (BC). **Methods**: This study was a cross-sectional study of adult H/L BC patients (n = 129). The participants were H/L women diagnosed with breast cancer (stages 0–4) in the past five years. Sociodemographic variables were assessed, as well as depression symptoms (Patient Health Questionnaire-9; Spanish version), grief (The Affective Neuroscience Personality Scales, Grief subscale; Spanish version), and general losses (Grief diagnostic instrument for general practice, loss categories section). **Results**: The mean age for the sample was 55.37 (SD = 11.57). The most frequent non-death-related losses were loss of liberty (f = 63, *p* = 48.8%), followed by fear of own death (f = 67, *p* = 51.9%) and loss of quality of life (f = 65, *p* = 50.4%). A higher mean rank was observed in depressive symptomatology scores for those who experienced loss of liberty (U = 73.91, *p* < 0.008), quality of life (U = 77.30, *p* < 0.001), and fear of their own death (U = 74.88, *p* < 0.002). The results indicate a significant positive relationship between the number of reported losses and depressive symptomatology (r = 0.340, *p* < 0.001). In terms of contextual factors, the participants who reported their income not being enough to cover their expenses reported a greater number of losses related to diagnosis (U = 74.67, *p* < 0.001) and more depressive symptomatology (U = 69.84, *p* = 0.041). Moreover, a relationship was observed between grief and academic levels (r = −0.234, *p* = 0.008). Likewise, a relationship was observed between age and the number of losses (r = −0.461, *p* < 0.001). **Conclusions**: Our results provide new evidence on how primary non-death-related losses due to a breast cancer diagnosis impact the mental health of H/L BC patients.

## 1. Introduction

Patients with cancer are at high risk of experiencing various types of losses due to their diagnosis and treatment. The burden caused by a cancer diagnosis, and/or treatment side effects may perpetuate losses such as financial loss, job loss, loss of quality of life, and changes in appearance (e.g., hair loss) [1,2,3,4]. Studies show that loss of social support leads to the onset of depression symptoms in breast cancer patients [5]. Additionally, loss of functionality could be distressing for patients with cancer. Some barriers associated with loss of functionality are fatigue and mobility problems, leading to low activity [6]. Loss of functionality can affect mental health, as higher functionality problems are associated with anxiety [7]. Furthermore, decreased self-efficiency has been associated with impaired mental health in cancer patients, underlying the importance of maintaining functionality [8]. Moreover, a symbolic loss, such as hair loss for women with cancer, has been shown to have adverse repercussions on mental health [3]. In a study, alopecia was consistently ranked as one of cancer treatment’s most distressing side effects [1]. Participants reported that hair loss disrupts how they experience their bodies, interact with others, and conceptualize their body image beyond treatment.

All these non-death-related losses can potentially lead to disenfranchised grief. Disenfranchised grief is a feeling of grief experienced after loss which is not normalized in our society [9], a concept that has been very little explored in patients with cancer. Grief has been widely conceptualized in psychology as a distressing feeling experienced after the death of a loved one, but there is also evidence suggesting that grief can be triggered by losses not related to the death of a loved one, such as loss of job, health, or perceived support [10,11,12,13]. However, the literature shows mixed findings regarding the interplay between non-death-related losses and grief, and this could be due to the use of models that are not empirically validated. Studies focusing on understanding non-death-related losses and mental health have broadly used models with poor research designs, consistency, and lack of empirical validation [14,15,16], which is why it is important to undertake a more holistic approach such as the biopsychosocial framework.

According to the affective neuroscience field, there are seven primary affects that compromise the functions of SEEKING-reward, RAGE, FEAR, sexual LUST, maternal and paternal CARE, separation-distress PANIC/GRIEF, and joyful PLAY [17,18]. The ultimate function of primary aspects is to help humans gather information about their surroundings for survival. One of these primary affects is the PANIC/GRIEF system. When the PANIC/GRIEF system is activated, studies in animal models have shown that rats make a call after separation to seek help and thus, prevent danger. The PANIC/GRIEF system is triggered by social separation, ultimately leading to despair [17,18]. Its primary function is to maintain social bonds as it conveys a sense of security [17,18]. Research on animal models has shown that the sustained and prolonged activation of this system inhibits the seeking-reward system and triggers depression symptoms [19]. Grief is often the immediate response to social loss, and researchers have linked the despair it brings to the reasons depression can be so painful [20]. The profound emotions felt during grief are understood as a form of psychological pain [21].

Nonetheless, there is a gap in knowledge regarding the expression of the PANIC/GRIEF system as conceptualized by affective neuroscience and its relationship with losses related to breast cancer diagnosis. Studies in the affective neuroscience field have shown that PANIC/GRIEF is associated with depressive symptomatology [22,23], yet, to our team’s knowledge, none have focused on H/L BC patients. The activation of the PANIC/GRIEF system is caused by social loss; thus, we hypothesize that other losses related to security and survival could also lead to grief, which could have a negative impact on the quality of life of patients. This is why the purpose of this manuscript is to assess non-death-related losses from a breast cancer diagnosis and its association with depressive symptomatology and grief in H/L breast cancer patients. The present manuscript has two aims: (1) to assess the relationship between losses related to cancer diagnoses, depressive symptomatology, and grief and (2) to assess contextual factors that impact losses related to cancer diagnosis, depressive symptomatology, and grief.

## 2. Materials and Methods

A cohort of 129 women with a BC diagnosis participated in this study. The described study is an ancillary study from a parent study titled “Biopsychosocial predictors of tumor-related inflammation and progression”, both funded by the United States of America’s (USA) National Institute of Minority Health and Health Disparities (NIMHD, U54MD). As part of the recruiting process, the research team contacted participants who provided written authorization to be included in a research recruitment registry from the parent study. Participant screening was performed via phone to confirm eligibility, explain the study, review the consent form, and answer any questions. Eligible individuals who consented to participate in the study were scheduled to complete the assessments by phone or face-to-face. The Ponce Research Institute Institutional Review Board (IRB) and Ethical Committee approved all the study procedures (Protocol 2205101801).

### 2.1. Participants’ Eligibility and Data Collection Process

This study sample consisted of women who identified as Hispanic/Latinas and were >21 years of age or older (adult legal age in Puerto Rico), and who were diagnosed in the past five years with breast cancer between the stages 0–4.

### 2.2. Measurements

All the outcome variables included in this study were assessed using the Spanish versions of each tool. First, a sociodemographic questionnaire was administered to collect data regarding the sample characteristics such as age, income, employment, academic level, and living situation. The following instruments were administrated for the data collection:

Patient Health Questionnaire-9 Spanish version (PHQ-9) [24]: The PHQ-9 is a self-report instrument that evaluates each of the 9 criteria of the DSM-IV diagnosis of depression. The Spanish version has shown a κ = 0.74 and overall accuracy, 88%; sensitivity, 87%; and specificity, 88%.

The Affective Neuroscience Personality Scales’ Spanish version [25]: For this study, only the SADNESS subscale was used. This subscale represents the concept of PANIC/GRIEF. The authors of this instrument designed the scale items to assess personal emotions and behaviors rather than cognitive and social judgment (e.g., “I tend to think about losing loved ones often” and “I often have the feeling that I am going to cry”). In an effort to reduce fatigue in our sample, we decided to choose the items from the SADNESS subscale of the ANPS Short version, which is composed of six items [26]. A reliability analysis was conducted for which we obtained a ω = 0.70, which indicates adequate reliability.

Grief diagnostic instrument for general practice’s loss categories section [27]. This instrument provides a checklist for various categories of losses (death-related and non-death-related). It provides a list of categories for the types of losses in a checklist form that the evaluator can provide to the subject (e.g., loss of a loved one, fear of death, migration or moving, etc.). It also provides a column with a list of examples to mention if the participant has trouble identifying losses that could fit the category. Additionally, it collects the time since the loss. For this study, the researchers modified the instructions to only assess the losses related to cancer diagnosis. To do this, we asked the participants to only check the losses they had that were related to their cancer diagnosis.

### 2.3. Data Analyses

Descriptive analyses, such as frequency and percentages analyses, were conducted to describe the sample’s sociodemographic and clinical characteristics, and to identify the most frequent losses related to cancer diagnosis. Mann–Whitney U non-parametric test was performed to assess which losses were more related to higher grief scores and depressive symptomatology, and to sociodemographic factors. Lastly, a Spearman Rho correlation test evaluated the relationship between the number of losses, depression, and grief, and sociodemographic factors. The study team used the IBM SPSS version 29 platform to conduct the analyses.

## 3. Results

### 3.1. Sociodemographic and Clinical Characteristics of the Sample

A cohort of 129 women with a BC diagnosis was utilized for these preliminary analyses. The following Table 1 describes all the sociodemographic and clinical variables collected in the sample. For numeric variables, mean and standard deviation were calculated. For categorical variables, frequency and percentage was calculated.

### 3.2. Non-Death-Related Losses Due to Breast Cancer

The most reported non-death-related loss due to a diagnosis of breast cancer was the fear of own death (f = 67, *p* = 51.9%), followed by loss of quality of life (f = 65, *p* = 50.4%) and loss of liberty (f = 63, *p* = 48.8%). Other non-death-related losses were financial loss (f = 37, *p* = 28.7%), loss of an opportunity (f = 34, *p* = 26.4%), job loss (f = 31, *p* = 24.0%), change in home (f = 19, *p* = 14.7%), partner or family separation (f = 18, *p* = 14.0%), and loss of integrity (f = 17, *p* = 13.2%). Lastly, a total of 24 (18.6%) patients reported other losses not mentioned in the checklist (e.g., loss of hair, breast, peace, and personal possessions).

### 3.3. Relationship Between Types of Non-Death-Related Losses, Depression, Grief, and Sociodemographic Factors

Out of the three most reported categories of losses, the team compared means for depression and grief to observe differences between groups, those who reported the loss vs. those who did not. A Mann–Whitney U non-parametric test was performed and showed a higher mean rank in depressive symptomatology scores for those who reported loss of quality of life (U = 77.30, *p* < 0.001), loss of liberty (U = 73.91, *p* < 0.008), and fear of own death (U = 74.88, *p* < 0.002). Likewise, it showed a higher mean rank in PANIC/GRIEF scores for those who reported the loss of quality of life (U = 71.22, *p* < 0.039). More, marginal significance was observed for loss of liberty (U = 70.98, *p* < 0.054). Regarding contextual factors, a Mann–Whitney U test indicates that the participants who reported being disabled presented more losses related to cancer diagnosis (U = 87.79, *p* = 0.015). Lastly, the participants who reported their income not being enough to cover their expenses reported a greater number of losses related to diagnosis (U = 74.67, *p* < 0.001) and more depressive symptomatology (U = 69.84, *p* = 0.041).

### 3.4. Relationship Between Number of Non-Death-Related Losses, Depression, Grief, and Sociodemographic Factors

To assess the relationship between the number of losses, depression, and grief, a Spearman correlation analysis was performed. A higher number of losses showed a significant positive correlation with depressive symptomatology (r = 0.340, *p* < 0.001). A positive correlation was observed between PANIC/GRIEF and depression symptoms (r = 0.502, *p* < 0.001). In terms of contextual factors, a Spearman analysis indicated a significant negative relationship between grief and academic levels (r = −0.234, *p* = 0.008). Likewise, a significant negative relationship was observed between age and the number of losses (r = −0.461, *p* < 0.001).

## 4. Discussion

The present study aimed to dissect the relationship between non-death-related losses, depressive symptomatology, grief, and contextual factors in H/L breast cancer patients. Our data showed that the cohort’s most commonly reported non-death-related losses were loss of freedom, quality of life, and fear of death. The patients in this study primarily reported the loss of freedom as the main non-death loss they experienced. Studies have stated that patients with cancer may experience existential issues by experiencing a loss of freedom of choice or liberty [28,29]. A patient might feel as if cancer has taken control of their life, and they lack control over it [30]. Moreover, patients may lose independence as they rely more on their caregivers [31,32], which can be associated with a loss of quality of life, the second most reported loss. This is an expected outcome, as Latina breast cancer patients also report worse quality of life than white breast care patients [33]. Moreover, breast cancer patients report deterioration in their quality of life, mostly resulting from cancer treatment side effects [34] that can lead to fatigue, insomnia, and menopausal symptoms [35].

Furthermore, our data showed how H/L breast cancer patients identify fear of death as a primary non-death-related loss. Fear of death has been associated with depressive symptoms [36], contributing to a lower quality of life and higher death anxiety [37]. The literature also supports the findings, as a higher mean rank for depression was overserved in those who reported fear of death compared to those who did not. Patients may experience fear of death due to their diagnosis and the physical consequences of diagnosis progression and treatment [38]. Fear of death can also be related to self-grief or preparatory grief. According to our data, higher PANIC/GRIEF scores indicated that the patients experienced more depressive symptomatology. The literature shows that cancer patients experience grief as they adapt to their cancer diagnosis, treatment side effects, and thoughts of death [39]. This preparatory grief has been associated with psychological distress [40].

Moreover, these non-death-related losses are associated with health loss or loss of functionality, which is impacted due to the cancer diagnosis, adjustment, and changes in the patient’s life. Depression is considered one of the main factors contributing to cancer patients’ functional status [41], where acute depressive symptomatology and quality of life affect an individual’s functionality in the physical, emotional, and social domains [42]. When faced with a diagnosis, cancer patients may experience preparatory grief due to the losses they encounter during treatment [43]. No significant relationship was found between the total number or types of losses and PANIC/GRIEF scores. However, the phenomenological components of grief must be considered. Culture could potentially play a role in the expression of PANIC/GRIEF scores. The literature shows that culture influences the interpretation of grief [44,45] as family dynamics, gender roles, values, and faith create an impact [46,47,48,49]. Particularly, our study sample was all women, and gender norms and roles lead to variations in how genders express and manifest grief [50]. Another important aspect of culture is religion, and the literature shows that religion can dictate appropriate grieving behavior [50]. Studies show that religion and spirituality are important aspects of how Latinos approach their cancer diagnosis [51].

Regarding contextual factors, the participants who reported their income not being enough to cover their expenses reported a greater number of losses related to diagnosis and more depressive symptomatology. This suggests that disadvantaged patients with low resources are positioned in an even more vulnerable situation, as they experience more losses than patients with a more stable socioeconomic status. According to the literature, cancer patients who experience financial distress report more intense physical and emotional symptoms [52]. Additionally, financial strain is linked to the severity of depression and anxiety symptoms [53]. Specifically, the financial burden associated with a cancer diagnosis is correlated with an increased likelihood of experiencing depressed mood [54]. Additionally, our data revealed a relationship between grief and academic levels. Higher grief was associated with lower academic levels. An explanation for this can be found in research indicating that higher education is linked to improved quality of life in cancer patients, which may indicate enhanced coping strategies [55]. Lastly, a significant negative relationship was observed between age and quantity of losses. Younger patients reported more losses. This could be because younger age is associated with a more productive phase in life, which may result in greater personal and professional investments, thereby increasing the stakes involved.

Future research can benefit from evaluating clinical factors that could impact the psychosocial variables evaluated in the present study. For example, it would be beneficial to assess how staging could affect grief and depression outcomes or its relationship with losses, as the literature has found staging to be directly associated with emotional distress [56]. Also, patients with metastasis have shown a higher risk of depression in comparison with patients with no metastasis [57]. Another factor to take into consideration is the time of diagnosis, as the literature has shown that depression symptoms increase when receiving a cancer diagnosis and then decline over time [58]. These results highlight the importance of taking into consideration contextual factors as they can play an important role when explaining variables such as loss, grief, and depression.

## 5. Conclusions

The results of this study bring a new perspective to the psycho-oncology field by focusing on the non-death-related losses experienced by H/L BC patients as part of their cancer experience, particularly by understanding the primary non-death-related losses that H/L breast cancer patients face during their diagnosis process and how these are impacting their mental health. Loss of freedom, quality of life, and fear of death are related to depression symptoms, putting breast cancer patients in a vulnerable situation. It also sheds light on the grief experience from an affective neuroscience point of view, confirming previous studies that observed a relationship between the PANIC/GRIEF system and depression symptoms. However, there is still a gap in knowledge surrounding the relationship between non-death-related losses and grief from a neuro-affective perspective, which could be addressed by validating the ANPS scale for the Puerto Rican population.

## Figures and Tables

**Table 1 healthcare-13-00624-t001:** Sociodemographic and clinical characteristics of the sample.

Sociodemographic Characteristics (n = 129)	Mean (SD)/f (%)
Age	55.37 (11.56)
Marital status	
Single	28 (21.7%)
Married or Consensual Union	71 (55.10%)
Divorced or Separated	18 (14.0%)
Widow	12 (9.3%)
Academic Level (High School or more)	
Did not finish elementary school.	1 (0.8%)
Elementary School (up to 6th grade)	4 (3.1%)
Middle School (up to 9th grade)	8 (6.2%)
High School (up to 12th grade)	30 (23.3%)
Technical Course/associate degree	27 (20.9%)
Bachelor’s Degree	44 (34.1%)
Master’s or Doctorate	15 (11.6%)
Employment	
Full-time or part-time	45 (34.9%)
Retired	24 (18.6%)
Unemployed	32 (24.8%)
Disabled	14 (10.9%)
Other	14 (10.9%)
Annual income	
≥USD 12,000	57 (44.2%)
USD 12,001–USD 19,000	26 (20.2%)
USD 19,001–USD 35,000	25 (19.4%)
USD 35,001–USD 60,000	16 (12.4%)
USD 60,001–USD 100,000	2 (1.6%)
USD 100,001–USD 250,000 a year	1 (0.8%)
Enough income to cover all expenses and needs	
Yes	51 (39.1%)
No	78 (60.9%)
Metastasis	
Yes	18 (14.1%)
No	111 (85.9%)
Treatment	
Surgery	100 (77.5%)
Chemotherapy	84 (65.1%)
Radiotherapy	69 (53.5%)
Hormone therapy	51 (39.5%)

## Data Availability

The data presented in this study are openly available on Figshare (https://doi.org/10.6084/m9.figshare.27868260.v1).

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
