# Peer review of "Losses Related to Breast Cancer Diagnosis: The Impact on Grief and Depression Symptomatology Within the Context of Hispanic/Latina Patients with Breast Cancer"

_healthcare, 2025, doi:10.3390/healthcare13060624_

Round 1
Reviewer 1 Report
Comments and Suggestions for Authors
Dear Authors,
This study highlights an important and often overlooked area—the losses experienced by cancer patients. I appreciate the authors' effort in addressing this topic. However, I believe the report can be improved with some revisions.
First, there is a grammatical error on line 36 that needs correction.
Since this study is based on the affective neuroscience field, the introduction should provide more detailed information about this approach. The theoretical explanation is not sufficient. The Panic/Grief component needs further clarification in the introduction. Since this study relies on this framework, providing more details would help the reader understand it better. In line with this, the section on disenfranchised grief can be shortened.
In the method section, it would be helpful to present some items from the Panic/Grief scale as examples. This would allow the reader to better understand the measurement process, including the number of items included.
There is also a lack of information in the measurement section regarding how the three loss-related items (loss of liberty, fear of death, etc.) were assessed and how these questions were asked. It would be helpful to clarify how these items were questioned.
In the results section, since socio-demographic characteristics are already provided in the table, repeating them in the text is unnecessary. The table is sufficient and explanatory.
Lastly, the discussion section should be reorganized to follow the flow of the findings. Presenting the correlation results at the end of paragraphs makes it difficult to follow. Therefore, Mann-Whitney U comparisons should be discussed after the first paragraph. The types of losses that showed higher scores for depression and grief should be discussed first. Then, the correlation analysis can be organized in a separate paragraph.
Author Response
Thank you for all your comments, observations, and suggestions. Your feedback will benefit our manuscript greatly as it will improve its quality and enrich its scientific value.
Comment 1: First, there is a grammatical error on line 36 that needs correction.
Response 1: This error has been corrected.
Comment 2: Since this study is based on the affective neuroscience field, the introduction should provide more detailed information about this approach. The theoretical explanation is not sufficient. The Panic/Grief component needs further clarification in the introduction. Since this study relies on this framework, providing more details would help the reader understand it better. In line with this, the section on disenfranchised grief can be shortened.
Response 2: Thank you for this observation. In the introduction, we have integrated more information regarding affective neuroscience theory. We hope the new information helps the readers integrate the information and understand better our hypothesis.
Comment 3: In the method section, it would be helpful to present some items from the Panic/Grief scale as examples. This would allow the reader to better understand the measurement process, including the number of items included.
Response 3: We have included items from the questionnaires to better understand the measurements.
Comment 4: There is also a lack of information in the measurement section regarding how the three loss-related items (loss of liberty, fear of death, etc.) were assessed and how these questions were asked. It would be helpful to clarify how these items were questioned.
Response 4: We have included more information regarding this survey and its format.
Comment 5: In the results section, since socio-demographic characteristics are already provided in the table, repeating them in the text is unnecessary. The table is sufficient and explanatory.
Response 5: We have eliminated the narrative section of the sociodemographic information.
Reviewer 2 Report
Comments and Suggestions for Authors
This study helpfully assesses the relationship in the study population between losses with a cancer diagnosis, depressive symptoms, and grief in the study population. It also assesses contextual factors impacting losses associated with a cancer diagnosis, depressive symptoms, and grief in the study population. The study provides helpful insights concerning non-death-related losses and grief associated with a breast cancer diagnosis. I recommend publication.
The study could be improved by additional attention in the Discussion and Conclusion to the crucial role of culture in the interpretation of grief issues. Does the impact of culture affect some subgroups in the study more strongly than others? Why might this be the case? Does the study suggest modifications or developments for current approaches to providing clinical support for patients with a breast cancer diagnosis? Does the study help to identify areas of concern relative to non-death-related losses and grief that are currently overlooked or deserve more attention?
Comments on the Quality of English Language
Although the quality of English language usage does not interfere with the clarity of the manuscript, the English language of this paper needs improvement. At line 49, "after losses" instead of "after a losses." At line 182, "participants who reported being disabled" (not "being disable"). At line 236, better to say "disadvantaged patients with low resources." Also at line 236, "an even." At 246 and 247, "number of losses."
Author Response
Thank you for your comments and recommendations. These will greatly improve te quality of our scientific paper.
Comment 1: The study could be improved by additional attention in the Discussion and Conclusion to the crucial role of culture in the interpretation of grief issues. Does the impact of culture affect some subgroups in the study more strongly than others? Why might this be the case? Does the study suggest modifications or developments for current approaches to providing clinical support for patients with a breast cancer diagnosis? Does the study help to identify areas of concern relative to non-death-related losses and grief that are currently overlooked or deserve more attention?
Response 1: We agree that the paper could benefit from explaining further the rule of cultural factors. You can refer to the discussion section to see additions. [224-232]
Comment 2: Although the quality of English language usage does not interfere with the clarity of the manuscript, the English language of this paper needs improvement. At line 49, "after losses" instead of "after a losses." At line 182, "participants who reported being disabled" (not "being disable"). At line 236, better to say "disadvantaged patients with low resources." Also at line 236, "an even." At 246 and 247, "number of losses."
Respose 2: These errors have been corrected and highlighted.
Reviewer 3 Report
Comments and Suggestions for Authors
The authors report evidence on symptoms (grief and depression) in Hispanic/Latina patients and very clearly described the need for data, methods and results.
Overall, the topic under investigation represents an interesting research area, but some topics require revision or discussion:
- Affiliation and correspondence data (Country, Department,…) is missing.
- reference numbers should be placed before the punctuation (e.g line 36)
- methods: please include a statement that data was prospectively assessed, if applicable
-Results: For a better overview, the authors should add a graphical representation of the symptoms in the results sectio
- The authors should add a paragraph to the discussion section on how the extent of breast cancer (stage 4, metastatic) as well as the extent of the applied, sometimes intensive therapy regimens, could have influenced their results
- The authors should add a limitation section to the discussion (e.g. breast cancer stage or the time between diagnosis and assessment of symptoms might also play an important role on the extent of assessed symptoms, …)
- Abstract: please use a colon instead of a period behind methods, results and conclusion
line 17: „women diagnosed with breast cancer stages 0-4, and in the past five years.“. please check the structure of the sentence.
line 36 „changes in appearance (e.g., hair loss)“ remove comma
line 64: remove comma after „one,“
line 84/85: please cite the reference, if the study has already been published
insert a space before the bracket (e.g. line 144,145, 149, Table 1 (annual income),….), line 181: remove space before „regarding“
line 158: report number „100 (77.5)“ behind „most participants“
Table 1: please add abbreviations to the subtitle (e.g. SD, f,…)
line 172+187 (heading 3.3. and 3.4): remove the statistical method from the titles
line 183: use lowercase for „Participants“
line 195: add a comma after „r=-.461“
line 236: please correct „dissanvangte“
Author Response
Thank you for your recommendations and observations. Your feedback is greatly appreciated as it will improve the quality of our manuscript.
Comment 1: Affiliation and correspondence data (Country, Department,…) is missing.
Response 1: The affiliations have been corrected and the information has been added.
Comment 2: Reference numbers should be placed before the punctuation (e.g line 36)
Response 2: All the reference numbers have been corrected.
Comment 3: methods: please include a statement that data was prospectively assessed, if applicable
Response 3: This does not apply to our study.
Comment 4: The authors should add a paragraph to the discussion section on how the extent of breast cancer (stage 4, metastatic) as well as the extent of the applied, sometimes intensive therapy regimens, could have influenced their results
Response 4: Thank you for this suggestion. We have added information regarding clinical factors and their impact.
Comment 5: The authors should add a limitation section to the discussion (e.g. breast cancer stage or the time between diagnosis and assessment of symptoms might also play an important role on the extent of assessed symptoms, …)
Response 5: In line with the preview comment. We integrated a paragraph discussing how future research should focus on taking into consideration some clinical variables that may impact the outcomes as our study did not include them
Comment 6: Please use a colon instead of a period behind methods, results, and conclusion
Response 6: This error was corrected.
Comments 7-17:
line 17: „women diagnosed with breast cancer stages 0-4, and in the past five years.“. please check the structure of the sentence.
line 36 „changes in appearance (e.g., hair loss)“ remove comma
line 64: remove comma after „one,“
line 84/85: please cite the reference, if the study has already been published
insert a space before the bracket (e.g. line 144,145, 149, Table 1 (annual income),….), line 181: remove space before „regarding“
line 158: report number „100 (77.5)“ behind „most participants“
Table 1: please add abbreviations to the subtitle (e.g. SD, f,…)
line 172+187 (heading 3.3. and 3.4): remove the statistical method from the titles
line 183: use lowercase for „Participants“
line 195: add a comma after „r=-.461“
line 236: please correct „dissanvangte“
Response 7-17: All of these errors have been corrected.